# EEG Dynamics of Error Processing and Associated Behavioral Adjustments in Preschool Children

**DOI:** 10.3390/brainsci13040575

**Published:** 2023-03-29

**Authors:** Marcos Luis Pietto, Federico Giovannetti, María Soledad Segretin, Sebastián Javier Lipina, Juan Esteban Kamienkowski

**Affiliations:** 1Unidad de Neurobiología Aplicada (UNA), CEMIC-CONICET, Buenos Aires C1431, Argentina; 2Laboratorio de Inteligencia Artificial Aplicada, Instituto de Ciencias de la Computación, FCEyN-UBA, CONICET, Buenos Aires C1428, Argentina

**Keywords:** ERN, frontal theta power, post-error slowing, post-error accuracy, go/no-go task, unsatisfied basic needs, preschool children

## Abstract

Preschool children show neural responses and make behavioral adjustments immediately following an error. However, there is a lack of evidence regarding how neural responses to error predict subsequent behavioral adjustments during childhood. The aim of our study was to explore the neural dynamics of error processing and associated behavioral adjustments in preschool children from unsatisfied basic needs (UBN) homes. Using EEG recordings during a go/no-go task, we examined within-subject associations between the error-related negativity (ERN), frontal theta power, post-error slowing, and post-error accuracy. Post-error accuracy increased linearly with post-error slowing, and there was no association between the neural activity of error processing and post-error accuracy. However, during successful error recovery, the frontal theta power, but not the ERN amplitude, was associated positively with post-error slowing. These findings indicated that preschool children from UBN homes adjusted their behavior following an error in an adaptive form and that the error-related theta activity may be associated with the adaptive forms of post-error behavior. Furthermore, our data support the adaptive theory of post-error slowing and point to some degree of separation between the neural mechanisms represented by the ERN and theta.

## 1. Introduction

The ability to detect errors and to use them to avoid future errors is central to everyday tasks. Errors allow us to learn and to adapt our behavior, cognition, and motivation to make our future actions more efficient. The detection and correction of errors are key components of cognitive control [1], and they are distinct aspects of performance monitoring [2]. The performance monitoring system proposed by Ullsperger et al. [3] detects when an error has been committed and signals the need for greater cognitive control to direct, to coordinate, and to engage lower-level processes to prevent the error from repeating [4,5].

Researchers generally use various tasks to study error processing (e.g., Flanker, Stroop, Simon, or go/no-go tasks) [6,7,8]. In these forced-choice tasks, an error happens when a person gives a response that does not match the response that should be associated with the present stimulus. When a person makes an error, behavioral adjustments are observed immediately following them. In the case of adult participants, compensatory adjustments may involve longer response times in trials that come after committing an error (known as post-error slowing or PES) [9], and improvements [1,10,11,12] or decrements [13,14,15,16] in accuracy on subsequent trials (known as post-error accuracy or PEA) may occur. A key question in the literature is whether PES represents an adaptive or maladaptive consequence of an error. Adaptive theories propose that this slowing is a strategic attempt (i.e., targeted along the sensory system, decision process, and/or motor initiation) to avoid future errors, which consequently facilitate a relative improvement in task performance (increased PEA) [1,17,18,19]. In contrast, maladaptive theories propose that this slowing instead reflects an impairment of cognitive processing [20,21], which potentially results in a deterioration of performance in subsequent trials (decreased PEA).

Following an error, specific electrophysiological responses were associated with the activation of the medial frontal cortex in adult participants [3,22,23,24,25,26]. The most studied neural response is error-related negativity (ERN), which is a response-locked, frontocentral event-related potential (ERP) component that peaks within 100 ms of an erroneous response [27,28]. It is thought to reflect the activity of either an error detection system that was sensitive to the mismatch between the overt response (the error) and correct response [27] or a conflict monitoring system that detects information discrepancies between the stimulus that provoked the activation of the correct response and the error just committed [1].

Theta-frequency oscillations (4–8 Hz) recorded over frontal cortical sites are another measure of neural responses following an error that occurs through increases in power. Whereas ERN is a specific electrophysiological response to errors, frontal theta oscillations are elicited by various task events that require increased cognitive control (e.g., conflict, punishment, and error). It is thought to act as a monitoring signal by which neurons calculate and communicate the need for control within and across broad brain networks [7] that are involved in cognitive control [29]. Frontal theta oscillations have been proposed to reflect the same neural processes as ERN, but they are manifested in a different domain [30]. However, recent evidence from adults has provided support for the notion that ERN and theta oscillations represented dissociable elements during error processing [10,12]. Thus, studying both measures separately may capture the relation between neural dynamics of error processing and post-error behavioral adaptations more comprehensively.

The control and regulation of voluntary behaviors become an increasingly important skill for children during preschool due to increased engagement in social and academic tasks with regulatory demands. During the first 5 y of life, children are subjected to rapid developmental changes in core cognitive functions that underlie performance monitoring and voluntary regulation of behavior (e.g., detection and correction of errors) [31,32]. Preschool children are less efficient than adults in monitoring and regulating their actions adaptively [33,34]. However, they engage the same neurocognitive control processes as adults to detect and to correct erroneous responses. In this sense, preschool children make behavioral adjustments and show neural responses following an error. For example, 3-year-old children have longer reaction times (RTs) following erroneous responses than correct ones [35,36,37,38], and they present neural responses to errors as increased ERN amplitude [6] and frontal theta power [39,40,41]. However, there is a lack of evidence regarding how neural responses to error (i.e., ERN and frontal theta) predict subsequent behavioral adjustments (i.e., PES and PEA) during childhood. In particular, we noted that while two studies [34,42] have provided some evidence to support a relationship between ERN and PES (larger amplitude–increased slow down) in children from 7–8 year old, there are also some studies [38,43] that have found null results on this relation in a sample with similar ages. Regarding PEA, three studies [34,38,42] showed a significant correlation between ERN and PEA in children from 5- to 8-year old, which suggested a role of ERN in predicting performance recovery during childhood. Finally, no studies have been found on the link between frontal theta oscillations and post-error behavioral compensation, despite growing evidence that suggested a key role of theta oscillations in orchestrating different cognitive control demands across preschool and school ages [33,44,45].

Most of the evidence about the predictive power of electrophysiological measures of error processing on PES and PEA came from adult studies. A meta-analytic review showed that larger PES was predicted to occur with larger error signals (i.e., ERN and frontal theta), and within-subjects analyses led typically to more consistent results than between-subject analyses [30]. However, recent within-subject evidence in adults suggested that frontal theta power was a stronger predictor of behavioral adjustments following an error (relative to the ERN amplitude), given that it predicted not only PES but also PEA [10,12].

The present study investigated electrophysiological and behavioral indices of error monitoring in preschool children during a go/no-go task. The sample involved in our study came from homes with low socioeconomic status (SES). Studies that investigated error processing in preschool showed that children from low-SES homes manifested neural dynamics of error processing and associated behavioral adjustments during go/no-go and conflict-related tasks [40,46,47,48].

However, there is a lack of evidence on how neural activity is linked to behavioral adjustments after errors in children from lower socioeconomic backgrounds, just as there is no such evidence for childhood in general. It is particularly important to address this gap in research because studies have indicated that the neural activity associated with error processing in children from lower socioeconomic environments may show distinct activation patterns [40] and developmental trajectories [46,47] compared with children from higher socioeconomic backgrounds. In our study, we did not compare children from different socioeconomic backgrounds. However, we focused on within-subject comparisons to better understand the mechanisms related to error processing in children from low SES. This approach may allow for a more detailed characterization of the neural activity associated with error processing in this population and provide insights into how socioeconomic context may affect cognitive functioning in children.

Specifically, we investigated the following points: (1) whether errors were associated with increased ERN amplitude and frontal theta power relative to correct responses, (2) whether there was an increase in response time on trials that followed an error (PES) and whether larger PES was associated with increases or decreases in subsequent trial accuracy (PEA), and (3) whether ERN and frontal theta oscillations predicted post-error behavior (i.e., PES and PEA). Based on previous results [35,36,37,38,39,41,46,49,50,51,52,53], we expected to find increased neural activity that followed errors relative to correct response in time-domain amplitude and time-frequency power. In addition, we hypothesized that reaction times would be slower on trials that followed errors relative to trials that followed correct responses and that PES would be related positively to PEA in support of adaptive theories of PES. Finally, we expected that frontal theta oscillations and ERN activity would affect post-error behavior positively (i.e., PES and PEA). 

## 2. Materials and Methods

### 2.1. Participants

A total of 120 children (64 males and 56 females, mean age ± *SD* = 5.3 ± 0.3 y) were recruited from a public kindergarten to participate in the study. Informed consent was obtained from parents or legal caregivers. The collection of data was conducted in 2017 and 2018. Inclusion criteria for participants included completion of more than one task block, EEG data with a reliable signal, and the presence of at least seven artifact-free error trials for the specific analysis.

The cutoff of seven artifact-free error trials was used because it is consistent with the cutoff recommended by other studies for EEG and behavioral error-related indices [12,54]. Exclusion criteria included history of prematurity or developmental disorders, failing to meet anthropometric standards [55], and the explicit reporting of exposure to childhood violence by mothers or school authorities. A total of 93 children (48 males and 45 females, mean age ± *SD* = 5.3 ± 0.3 y) were involved in the analyses. The sample size was decreased due to the following reasons: (a) the presence of developmental disorders (*N* = 3), (b) no completion of a single task block (*N* = 4), (c) EEG signal was contaminated by the presence of excessive artifacts (*N* = 20), and (d) presence of fewer than seven artifact-free error trials (median = 35). Although we excluded children from the sample in a pre-analytical phase, all children were authorized to participate in the study activities.

### 2.2. Sociodemographic Scale

To obtain data on sociodemographic characteristics, family household information, and child health conditions, we administered a questionnaire [56] to one parent or caregiver of each child. The variables included in the present study involved unsatisfied basic needs (UBN) indicators. We analyzed the presence of each UBN indicator in the sample and the number of subjects from households with UBN with a specific number of indicators (see Appendix A for details).

### 2.3. Task

Participants completed a child-friendly version of the go/no-go task [48]. Each trial began with the presentation of a stimulus (either a go or a no-go) that was displayed for 400 ms. It was followed immediately by a black cross that remained visible for 800 ms and had a visual angle of 2°. The go condition was presented in 70% of the trials, and the remaining 30% involved the no-go condition. Children were instructed to respond as quickly as possible to each go stimulus by pressing the “space” key, but not to respond to no-go stimuli (Figure 1). The stimuli did not disappear following the button press. The stimuli were displayed on a laptop monitor with a screen size of 15.6 inches, a screen resolution of 1366 × 768 pixels, and a refresh rate of 60 Hz. Responses from the participants were collected using a standard keyboard. During the experiment, the computer was positioned 40 cm away from the child.

All the stimuli used in the study were created using the PsychoPy toolbox (version 3.0) for Python (version 2.7). The stimuli were images of characters from popular video games such as Pacman and Angry Birds. Specifically, four types of no-go stimuli and one type of go stimulus were used. The go stimulus was represented by the Pacman/Bird character, and the no-go stimuli were represented by the Ghosts/Pigs characters. The images had five different body colors defined by their RGB values. Specifically, the Pacman/Bird character was represented by the color yellow (RGB values: 253, 217, 47), and the Ghosts/Pigs characters were represented by four different colors: Blue (RGB values: 47, 140, 253), Green (RGB values: 48, 253, 72), Orange (RGB values: 253, 135, 48), and Magenta (RGB values: 250, 47, 253). All stimuli were presented at the center of the screen on a gray background (RGB values: 150, 150, 150) and occupied a visual angle of 8.84 degrees horizontally and vertically. Each participant completed a minimum of three blocks and a maximum of eight blocks. The number of blocks completed by the children varied between three and eight because they could finish the task before completing all eight blocks if they were fatigued or chose not to proceed. Each block consisted of 90 trials (720 trials in total), which were distributed in two sections of four blocks. The first section involved the Pacman characters, and the second section involved the Angry Birds characters. The information was collected after exposing children to > 20 trials of practice.

All behavioral measures were estimated including only trials with go (i.e., hits) or no-go (i.e., false alarms) behavioral responses. Error-trial RT and correct-trial RT were computed as response time to no-go stimuli (i.e., false alarms) and go stimuli (i.e., hits), respectively. PES and post-correct slowing (PCS) were calculated at a single-trial level by subtracting the current trial RT (e.g., Error-trial RT) from the next trial RT (e.g., post-error RT). PCS trials were RT-matched randomly to PES trials to control for differences in RT due to regression toward the mean. This procedure was repeated 20 times. At each one, we computed the mean of the PCS for each subject and analysis and finally we performed the average of the 20 means. PEA and post-correct accuracy (PCA) were computed as the percentage of behavioral responses associated with hits following a false alarm and hit, respectively. The PCA trials were also matched to the PEA trials using the same random procedure performed for RT.

### 2.4. Electrophysiological Recording and Processing

We recorded neural activity using the Emotiv EPOC+ system (www.emotiv.com, accessed on 26 March 2023), which contained 14 saline-based electrodes positioned on the scalp according to the International 10–20 system. The EEG signal was digitized at 128 Hz and was bandpass filtered between 0.16 Hz and 43 Hz during the recording. We measured impedances before and during recording and kept them according to EPOC calibration at the threshold between yellow and green.

The same python program executed the EEG recordings and the visual presentation of the go/no-go task through functions reported in [57] (https://github.com/mathigatti/Emotiv-Experiments, accessed on 26 March 2023). The processing and analysis of EEG data were performed using the EEGLAB (version 13.5.4b) MATLAB (version R2016a) toolbox.

#### 2.4.1. Event-Related Potentials (ERPs)

We applied a bandpass filter using a finite impulse response filter over the frequency range from 0.5 to 30 Hz to each participant’s EEG data. We then segmented the continuous EEG data into epochs of 450 ms before to 550 ms after stimulus response in trials related to correct (i.e., response to go stimuli–hits) and error responses (i.e., response to no-go stimuli–false alarms). The baseline activity was subtracted from each epoch using the average activity in the interval [−450, −50] ms. We used the independent component analysis (ICA) to remove artifacts related to eye movements, blinks, and signal noise. The remaining artifacts were rejected automatically from epochs with voltage fluctuations that exceeded ±110 μV. We re-referenced the response-locked signals to a pseudo-average reference across symmetrically distributed electrodes around the scalp, which included the channels O1, O2, T7, T8, AF3, and AF4. Finally, we computed the average activity over a frontal region of interest (ROI) that consisted of F3/F4 electrodes. 

We used a denoizing algorithm to obtain a clean single-trial ERP [57]. As shown previously [58,59,60], this method improved the estimation of waveforms compared with the non-denoized, single-trial waveforms significantly. The response-locked activity was reconstructed by averaging clean trials for each participant and condition [57].

#### 2.4.2. Event-Related Spectral Perturbations (ERSPs)

The EEG data processing for the ERSPs was similar to the one used for the ERPs. It was completed using the following order: (1) 0.5–30 Hz bandpass filtering. (2) ICA components removal. (3) 1 Hz band filtering: the signal was filtered over the 1–30 Hz range using filters of 1 Hz sizes. (4) Signal re-referencing. (5) Envelope calculation from the Hilbert Transform: we applied it to extract the instantaneous power values over each signal. For theta-band analysis, power data were calculated for the frequency range from 4 to 7 Hz. (6) Signal segmentation into epochs. (7) Baseline normalization: we used a decibel-normalized power. As Cohen [61] recommended, the baseline normalization was applied on the trial-average power. We first averaged trials and then transformed them into decibels using the average activity in the interval [−450, −350] ms. (8) Epoch rejections. (9) Frontal ROI computation.

All analyses were performed on the response-locked activity at the frontal ROI F3/F4. The reason for this choice was based on the topography and effects of error-related signals reported in previous studies [8,38,41,43,46,48,62]. We computed the response-locked activity (amplitude and theta power) from error- and correct-related responses. In addition, we calculated the response-locked activity of errors that were followed by correct responses (successful error recovery) and incorrect responses (failed error recovery). We included only error trials followed by hits for the successful error recovery condition, but for the failed error recovery condition, we included errors followed by false alarms.

### 2.5. Data Analysis

We used parametric tests to compare neural activity variables and nonparametric tests for the rest of the variables because normality and/or homoscedasticity were not met (see Appendix A for details).

The cutoff of seven artifact-free error trials was used because it is consistent with the cutoff recommended by other studies for the EEG and behavioral error-related indices [46,54]. The low number of error trials imposed a limitation on the experiment because it reduced the number of double error trials (i.e., a false alarm followed by another false alarm). This could be related to a low false alarm rate (~36%) and to the random trial distribution (low probability of presenting two consecutive no-go trials) (see Appendix A for details).

#### 2.5.1. Behavior

In line with the existing literature [38,43,46], our initial prediction was that responses associated with errors would be faster than those associated with correct responses. Thus, we analyzed whether the behavior associated with an error differed from a correct response. Specifically, error-trial RTs were compared with correct-trial RTs using the Wilcoxon signed-rank test.

Second, we predicted that trials that followed errors would result in slower reaction times and increased accuracy compared with trials that followed correct responses. To this end, we tested whether there was a difference in behavioral adjustments that followed an error compared with those that followed a correct response. Specifically, PES and PEA were compared with PCS and PCA, respectively, using the Wilcoxon signed-rank test.

Our final prediction was that there was a positive relationship between PES and PEA, in line with adaptive theories of PES. To this end, we examined whether the RT slowing that followed an error (i.e., PES) was related to subsequent trial accuracy (i.e., PEA). Specifically, the PEA of each subject was divided into tertiles based on its PES. We used the Friedman test to evaluate differences in accuracy across the multiple PES levels and the Wilcoxon signed-rank test to perform pairwise, post hoc comparisons.

#### 2.5.2. Neural Activity

Our first expectation was that there would be a greater neural activity in the time-domain amplitude and time-frequency power that followed errors compared with correct responses. For that purpose, response-locked activity (amplitude and theta power from error- and correct-related responses) was compared using non-parametric, permutation *t*-tests for each sample along [−50 ms, 150 ms] and [−350 ms, 450 ms] time windows for the ERP and ERSP, respectively.

Then, we predicted that frontal theta oscillations and ERN activity were associated positively with post-error behavior, namely PES and PEA. To evaluate whether neural activity was associated with PEA, we compared the response-locked activity of errors that were followed by correct responses with those followed by incorrect responses through the same procedure described above. To examine whether neural activity was related to PES, we compared neural activity before an error recovery (i.e., hit that follows a false alarm) across high and low levels of PES. Specifically, each subject’s error-related activity (i.e., amplitude and theta power) that was followed by a hit was divided into tertiles based on its PES. We first computed the area of response-locked waveforms within a specified time-window: [−50 ms, 150 ms] and [−350 ms, 450 ms] for the ERP and ERSP, respectively. Then, we used a one-way ANOVA to evaluate differences in neural activity area across the multiple PES levels, and we used a paired sample *t*-test to perform pairwise post hoc comparisons. Finally, we computed the effect sizes for all comparisons using *r*-values [63].

#### 2.5.3. Response–Stimulus Interval Analysis

Response–stimulus interval (RSI) refers to the amount of time between the participant’s response to a stimulus and the presentation of the next task stimulus. The length of the RSI that followed an error response influenced PES and PEA in both adults [10,14] and children [37]. Therefore, we examined whether the amount of time before the next task-relevant stimulus that followed an error (i.e., RSI) was different across PES tertiles. To this end, we compared the RSIs across the PES tertiles within the samples used for behavior and neural levels. We used the Friedman test to evaluate differences in the RSIs across the multiple slow-down levels, and we used the Wilcoxon signed-rank test to perform pairwise post hoc comparisons.

## 3. Results

### 3.1. Sociodemographic Characteristics of the Sample

This analysis revealed that 96.4% of the children in our sample came from homes with UBN. The sociodemographic characteristics of these families showed that 95.2% of them lived in disadvantaged neighborhoods. Additionally, 46.5% of the families exhibited two or more UBN indicators (Table 1).

### 3.2. Behavior

The Error-trial RT was shorter than correct-trial RT (Mdn = 481.2 ms compared with Mdn = 587.4 ms, *Z* = −7.99, *p* < 0.001, *r* = 0.83) (Table 2). After subtracting the current trial RT from the next trial RT, we observed that the RT slowing that followed an error was higher than the RT slowing that followed a correct response (Mdn = 148.3 ms compared with Mdn = 10.3 ms, *Z* = 8.03, *p* < 0.001, *r* = 0.84), which confirmed the presence of PES in the task. On the other hand, the PEA was not significantly different from the PCA (Mdn = 83.6% compared with Mdn = 83.5%, *Z* = 1.43, *p* = 0.152, *r* = 0.15).

We performed a within-subject analysis to test whether the slow down that followed the error was accompanied by increased accuracy in the subsequent trial. Each subject’s error trials, whether followed by a correct or an error response, were grouped into tertiles based on PES. Sixty-six participants with at least seven error trials in each of the three tertiles were included in these analyses. The tertile medians of PES were −125.7 ms, 119.5 ms, and 429.5 ms. There were significant differences in PEA between tertiles (*χ*^2^ (2) = 17.68, *p* < 0.001), in which accuracy increased significantly as a function of PES (Figure 2). Post hoc comparisons revealed that the 3rd tertile had a significantly (significance level after Bonferroni correction *p* = 0.0167) greater accuracy relative to the 2nd tertile (*Z* = 2.75, *p* = 0.006, *r* = 0.34) and the 1st tertile (*Z* = 4.42, *p* < 0.001, *r* = 0.54). Morever, the 2nd tertile had a significantly greater accuracy relative to the 1st tertile (*Z* = 2.45, *p* = 0.014, *r* = 0.30).

### 3.3. Electrophysiology

We conducted non-parametric bootstrap permutation *t*-tests to evaluate if errors in the go/no-go task elicited an ERN. We contrasted each sample along the ERP time-window from −50 to 150 ms by comparing the error and correct trials. Ninety-three subjects with at least seven error trials were included in this analysis. Permutation analysis revealed differences in amplitude between error and correct response waveforms (Bootstrapping: minimum and maximum *t* values = 1.63–6.30, *p* < 0.01; time-window: [5 ms, 145 ms]; Figure 3). In particular, the ERN amplitude was significantly larger (i.e., more negative) during the error trials than during the correct trials. A similar permutation test was performed on the time course of the ERSP data to evaluate if errors in the task also elicited a theta activity. The error and correct trials were contrasted along a time-window from −350 to 450 ms. Ninety-three subjects were included in this analysis. There was a significantly greater theta power following the errors than following the correct responses (Bootstrapping: minimum and maximum *t* values = 1.79–8.21, *p* < 0.01. time-window: [−277 ms, 442 ms]; Figure 3).

To examine whether the error-locked waveforms involved different neural activity depending on whether it was followed by a correct or an error response, non-parametric bootstrap permutation *t*-tests were performed on the ERP and ERSP data. Forty-five subjects with at least seven error trials were included in this analysis. There were no significant differences in amplitude (Bootstrapping: minimum and maximum *t* values = −0.58–0.72, all *p* > 0.246, time-window: [−50 ms, 150 ms]) and theta power (Bootstrapping: minimum and maximum *t* values = −0.10–0.93, all *p* > 0.140, time-window: [−350 ms, 450 ms]) of the response-locked activity between the successful error recovery and the failed error recovery conditions (Figure 4).

To investigate the relationship between the ERN, error-related theta activity, and PES during successful error recovery, we repeated the procedure of grouping the error trials in tertiles. However, this time we only included the error trials that were followed by the correct responses. Fifty-eight subjects with at least seven error trials in each of the three tertiles were included in these analyses. The tertile medians for PES were −119.5 ms, 121.7 ms, and 428.6 ms. There was a statistically significant difference between the tertiles in the error-related theta (*F*(2,171) = 15.90, *p* < 0.001), but not in the ERN (*F*(2,171) = 0.26, *p* = 0.772). In particular, a higher PES was related to a greater theta power during successful error recovery (1st tertile—Lower PESMean = 45.6, 2nd tertile Mean = 56.8, 3rd tertile—Higher PESMean = 89.7; see Figure 5). Post hoc comparisons revealed that the 3rd tertile had a significantly greater theta power relative to the 2nd (*t*(57) = 4.38, *p* < 0.001, *r* = 0.58) or the 1st tertile (*t*(57) = 5.12, *p* < 0.001, *r* = 0.67) (Bonferroni correction *p* = 0.0167). However, the 2nd tertile did not have a greater theta power relative to the 1st (*t*(57) = 1.36, *p* = 0.180, *r* = 0.17). As shown in Figure 6, this power difference between errors that preceded the correct trials of the different PES levels had a frontal topography.

### 3.4. RSI

There were significant differences in the RSIs between the tertiles used for behavior (*χ*^2^ (2) = 120.27, *p* < 0.001). The time before the subsequent task-relevant stimulus increased significantly as a function of the PES (1st tertile—Lower PESMdn = 597 ms, 2nd tertile Mdn = 783 ms, 3rd tertile—Higher PESMdn = 898 ms). Post hoc comparisons revealed that the 3rd tertile had a significantly greater RSI relative to the 2nd (*Z* = 6.91, *p* < 0.001, *r* = 0.85) or the 1st tertile (*Z* = 7.06, *p* < 0.001, *r* = 0.87) (significance level alter Bonferroni correction *p* = 0.0167). Moreover, the 2nd tertile had a significantly greater RSI than the 1st tertile (*Z* = 7.02, *p* < 0.001, *r* = 0.86).

As in the case of behavior, the RSIs between the tertiles used for neural activity were also significantly different (*χ*^2^ (2) = 112.14, *p* < 0.001). A greater PES before successful error recovery was related to a longer RSI (1st tertile—Lower PESMdn = 578 ms, 2nd tertile Mdn = 766 ms, 3rd tertile—Higher PESMdn = 886 ms). Post hoc comparisons revealed that the 3rd tertile had a significantly (significance level alter Bonferroni correction *p* = 0.0167) greater RSI relative to the 2nd (*Z* = 6.62, *p* < 0.001, *r* = 0.87) or 1st tertile (*Z* = 6.62, *p* < 0.001, *r* = 0.87). Moreover, the 2nd tertile had a significantly greater RSI relative to the 1st (*Z* = 6.59, *p* < 0.001, *r* = 0.87).

## 4. Discussion

We investigated the neural activity of error processing (as indexed by the ERN and frontal theta), post-error behavioral adaptations, and the relationship between them in a sample of preschool children from UBN homes. We used a series of within-subject comparisons to test whether a greater PES was associated with an increased PEA and whether the frontal theta power and ERN amplitude were associated with PES and PEA. At the behavioral level, PEA increased linearly with PES, which suggested a role for PES in successful error recovery. Then, we found a greater frontal theta power and a larger (more negative) ERN amplitude after the error responses than after the correct responses. Moreover, we did not find an association between the neural activity of error processing and PEA. However, during successful error recovery, the frontal theta power, but not the ERN amplitude, was associated positively with PES.

Our findings on the presence of neural dynamics of error processing and associated behavioral adjustments were consistent with past studies based on preschool children during a go/no-go [38,39,41,46,49,50,51,52,53] and other cognitive control tasks [35,36,62,64,65]. In addition, our results were consistent with previous work that showed neural responses to error and post-error behavior in preschool children from low-SES homes [40,46,47]. To our knowledge, this study is one of the first to investigate the relationship between the neural activity of error processing and subsequent behavioral compensation in children from UBN homes. Furthermore, we note that while some studies have examined the relationship between the ERN and post-error behavior [34,38,42,43], no studies have employed time-frequency measures, such as the frontal theta, to evaluate its relationship with PES and PEA.

In accordance with recent studies that were implemented during childhood [42], the within-subject analysis revealed that PES was associated with PEA. We found that PEA increased linearly with PES, which provided further evidence that PES predicted performance recovery. This finding supports the adaptive theory of PES, which postulates that slowing down after an error reflects a more cautious response strategy to increase task performance. In the literature, however, PES was associated less frequently with PEA. A possible explanation for these findings could be that PES is not a unitary construct, but may reflect different processes that depend on the RSI (i.e., the amount of time before the next task-relevant stimulus following an error) [14].

An emerging view [5] is that error induces a cascade of cognitive events that must be finished before the presentation of a subsequent stimulus to enable adaptive improvements on the next trial. From this perspective, events are represented by an automatic and general early stage, in which there is an inhibition of the cognitive system, an attentional orienting response to error, and a late stage of controlled and adaptive error processes that yield an improvement in accuracy on the next trial. At short RSIs, the presentation of task-relevant stimuli overlapped early or late stages of error processing. Consequently, PES results from a necessary reorienting of attention toward the initial task set, and no increased PEA would be observed. On the contrary, automatic and controlled post-error processes finished at long RSIs so that the PES resulted from a strategic and deliberative speed-accuracy trade-off. In sum, PES may be related positively to PEA whether or not there is sufficient time before the trial presentation that follows an error. Our results are generally consistent with this hypothesis, given that the RSI increased as a function of the PES tertiles. In other words, trials associated with a greater PEA involved a longer RSI than trials with a smaller PEA. Thus, increased PEA was related to a larger RT slow down after an error and to more available time to adapt behavior until the next response was performed.

When investigating whether neural dynamics of error processing were related to PEA, within-subject comparisons revealed that the theta power and ERN amplitude did not differentiate error that was followed by a correct response from an error that was followed by another error response. Our results in ERN contrasted with those studies that were implemented with children and adolescents 5–18 year of age [34,38,42], which provided some evidence in support of a relation between ERN and PEA. Unlike our study, Kang et al. [34] used between-subject analyses and showed that the significant correlation between ERN and PEA was shared with the age effect between children and adolescents. Moreover, Overbye et al. [42] and Torpey et al. [38] found only significant correlations when they computed both PEA and/or ERN as difference measures, which were calculated by subtracting one measure from another. Furthermore, Overbye et al. [42] observed that the associations between ERN and PEA in children and adolescents 8–19 year old were weaker at younger ages, which indicated a reduced ability of neural indices of error processing to predict and to guide behavior during childhood. Moreover, we also did not observe a significant association between the theta power and PEA. Although to our knowledge no studies have examined the relationship between the theta power and PEA in children, recent studies have provided evidence for a relationship between the frontal theta power and PEA in adults [10,12]. Note that the sources of between-study variability may also be related to the younger age (i.e., 5 y old compared with 8 year old) and the higher proportion of children from UBN households in our sample than in the studies mentioned above.

Importantly, we observed that the theta power was associated positively with PES before successful error recovery. More specifically, when an error response was followed by a correct response (i.e., a hit), a higher theta power on the current trial was associated with a greater RT slowing on the subsequent trial. We also observed that longer RSIs drove trials with a greater slowing following an error than those with smaller RT slowing. Therefore, as the time to process an error increased, we observed a greater theta activity, which resulted in an increase in response times on the following post-error trial. Given that frontal theta oscillations are generally elicited by task events that require increased cognitive control [7,45], the involvement of controlled processes underlying error recovery could have been different across the PES tertiles. When we compared the magnitude of theta oscillations between the PES tertiles, only the highest PES tertile had a greater theta power than the middle and lowest PES tertiles.

One possible explanation for this pattern of results is that the highest PES tertile was more likely to reflect an adaptive or genuine recovery from an error compared with the lowest and middle PES tertiles. Contrary to the lowest and middle PES tertiles, the highest PES also involved longer RSI [14], that is, an additional time to process errors correctly before subsequent presentation of trial stimuli. It has been hypothesized that controlled processes are implemented chronologically later than automatic post-error processes when there is a long time between the error response and the subsequent presentation of trial stimuli [5]. Therefore, recovery from error at a greater PES and longer RSI could have captured deliberated and controlled processes that enabled a strategic compensation of behavior. Conversely, a correct response that followed an error at smaller PES and shorter RSI could have indicated no real behavior correction, but rather a perseverative automatic motor response from the previous error trial.

The presence of trials with non-genuine error correction within the error recovery condition might also explain why error-related theta was not differentiated between successful error recovery compared with failed error recovery. Error-related trials with subsequent perseverative motor responses (i.e., errors) engage similar cognitive control demands. If this was the case, we expect similar theta magnitudes independently if they resulted in a successful or failed error recovery. In this sense, we observed that the power of error trials in the lowest and middle tertiles of PES that preceded a correct response (i.e., a hit) was not significantly greater than the power of trials that preceded another error response (i.e., a false alarm) (see Appendix A for details). This finding suggested that error trials with smaller RT slowing and shorter available time to adapt behavior before a correct response did not involve greater cognitive control demands than error trials followed by another error response. Instead, the error-related theta power in the highest tertile of PES that preceded a correct response (i.e., a hit) was significantly greater than the power of trials that preceded another error response (i.e., a false alarm). These results suggested there was an increased recruitment of signals, which indicated the need for cognitive control to recover from an error only at a greater PES and longer RSI. This finding is consistent with previous evidence in children [34,38,42], where a relationship between the neural activity of error processing and PEA was observed at longer RSI.

In addition, we observed that time-frequency power, but not the ERP amplitude, was related to post-error behavior, given that theta, and not ERN, increased as a function of PES before an error recovery. This result was concordant with recent evidence in adults that supported the notion that frontal theta and ERN had different abilities to predict post-error behavioral adjustments and had separate roles during performance monitoring. [10,12]. In neurophysiological terms, time-frequency-domain averaging (e.g., theta power) captures a different portion of dynamic and multidimensional space of brain processing than time-domain averaging (i.e., ERPs). Whereas both EEG signals measure activity time-locked to errors that include transient phase-locked activity (or evoked signal), time-frequency power also captures oscillatory and non-phase-locked neural activity (or induced signal). In this sense, there is evidence that ERN emerges from phase locking of frontal theta activity. However, following errors, the increase in the non-phase-locked theta power was larger and longer-lasting than phase-locked theta power [23,66]. In addition, Beatty et al. [10] recently showed that the non-phase-locked theta power was a more robust predictor than ERN, given that it related positively to PES and PEA. These results suggested that time-frequency theta oscillations related more closely to slow and controlled processes that served adaptive forms of post-error behavior than ERN.

In our analyses, we included subjects who had at least seven trials in each condition to obtain reliable measures. It is important to note that these estimates were derived from an adult sample [54], and it is possible that reliability may vary more in samples of preschoolers due to the fact that development is associated with increases in variability in neural activity [67]. Future studies will be crucial in determining whether more reliable neural activity of error processing distinguishes and reflects behavioral indices of error monitoring more accurately.

The cutoff of seven trials imposed a limitation on the experiment because it reduced the sample size considerably, especially when we considered double error trials (i.e., a false alarm followed by another false alarm). On the one hand, the presence of few double error trials was because the children did not commit numerous false alarms (~36%). On the other hand, this was also due to the paradigm design. The paradigm was designed to show trials randomly with a 70% probability of presenting go trials and a 30% probability of presenting no-go trials. Thus, the probability of presenting two consecutive no-go trials was lower than the other possible combinations (no-go/no-go = 9%; no-go/go = 21%; go/no-go = 21%; go/go = 49%).

We might address this methodological artifact in various ways in future research. First, we could increase the task’s difficulty to elicit a large percentage of false alarms. However, involving children in a more difficult task might expose them to frustration or other unnecessary negative feelings. In addition, changing the relative frequency of false alarms could affect the value of errors (from unexpected to expected events) and, thus, subsequent behavioral adjustments [20]. Second, we could include more trials in the task. Instead, this leads to a longer experimental session. We should extend the experimental session without increasing the probability of inducing participant fatigue. Third, we could select the same number of trials, but order them differently. For example, trials could be distributed in a pseudo-casual manner so that there are a reasonable number of consecutive no-go trials in each experimental session.

This study was implemented in a sample of kindergartners from UBN homes. While this approach enabled us to characterize error monitoring at the neural and behavioral levels of analyses in a low-SES sample, its limited socioeconomic variability prevented us from exploring potential associations between error processing and the number of UBN indicators. To better understand the contextual factors that may influence error processing, it is crucial to address this issue in future studies. By examining the relationships between different numbers of UBN indicators and error monitoring in a more diverse sample, we would be able to explore more specifically the eventual modulator role of distinct SES-related contextual factors [68].

Our study was conducted in an educational setting, which has been shown to have advantages in previous studies [69,70]. However, the study of cognitive processes from different levels of organization (e.g., neural, cognitive) is not typically implemented outside of laboratory settings due to the added noise and logistical challenge. To accomplish this, we utilized mobile EEG technology, and we found that the results aligned with previous EEG research on performance monitoring processes [35,36,38,39,41,46,49,50,51,52,53,62,64,65]. Thus, the data from our study indicated that it is feasible to apply electroencephalographic evaluation methods outside of the laboratory setting using mobile EEG technology. Nevertheless, we recognize a constraint in using mobile EEG technology with a limited number of electrodes because this montage may not capture electrophysiological signals accurately across the entire scalp.

Finally, the findings of our work indicated that preschool children from UBN homes adjust their behavior following an error in an adaptive form and that the error-related theta activity may be associated with adaptive forms of post-error behavior. Furthermore, our data provide evidence for the adaptive theory of PES [5] and highlight partial dissociations in the neural mechanisms captured by the ERN and theta.

## Figures and Tables

**Figure 1 brainsci-13-00575-f001:**
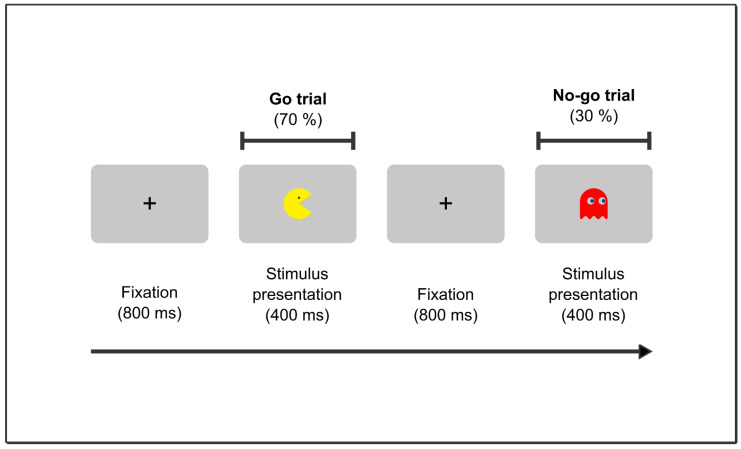
Design of the go/no-go paradigm. Stimuli size does not reflect the one used in the experiment. They have been changed for presentation purposes.

**Figure 2 brainsci-13-00575-f002:**
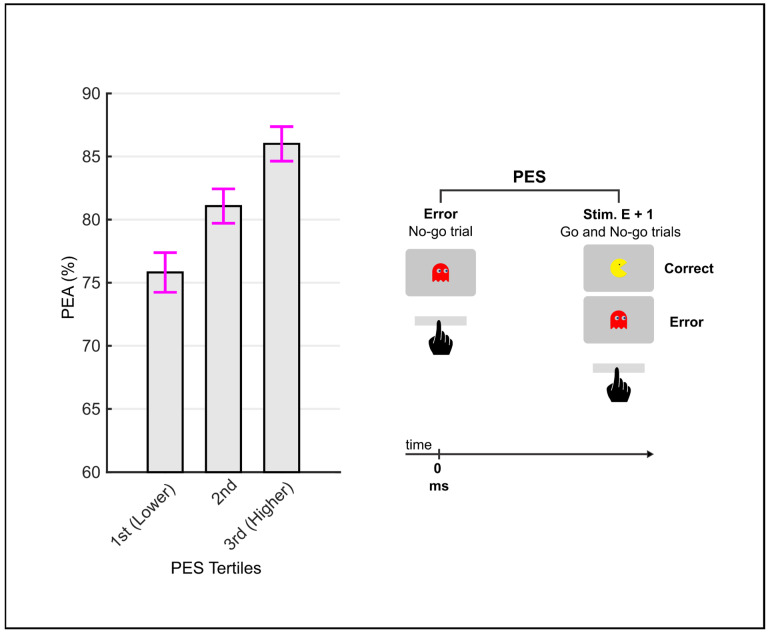
Post-error accuracy (PEA) as a function of post-error slowing (PES). Error bars depict standard errors. There were significant differences in PEA between tertiles (*p* < 0.001), in which accuracy increased significantly as a function of PES.

**Figure 3 brainsci-13-00575-f003:**
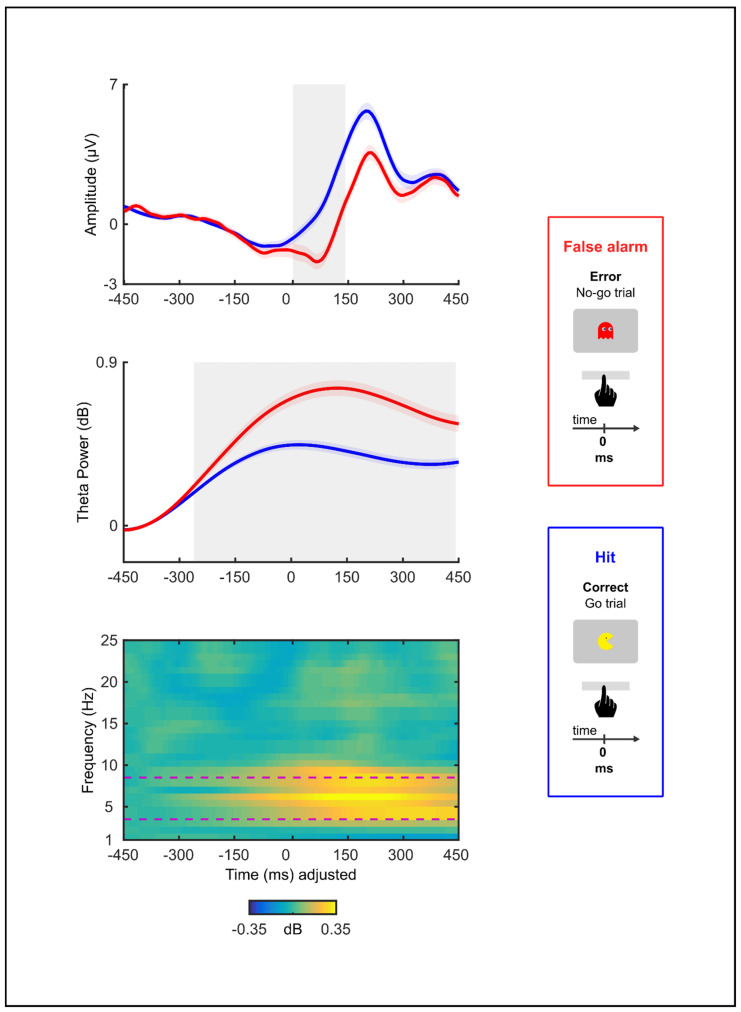
Grand averages of response-locked activity from error trials and correct trials (F3/F4) (n = 93). The top graph shows time-domain amplitude. The middle graph shows temporal frequency powers in the theta band. The bottom graph shows temporal frequency powers after subtracting correct trials from errors in the frequency range from 1 to 25 Hz. The button press occurred at 0 s. Shaded areas represent significant differences (*p* < 0.01) in amplitude and power in ≥5 samples (~40 ms). The response-locked activity was compared using non-parametric, permutation *t*-tests for each sample along [−50 ms, 150 ms] and [−350 ms, 450 ms] time windows for time-domain amplitude and time-frequency power, respectively. Permutation analysis revealed differences in amplitude (shaded area: [5 ms, 145 ms]) and theta power (shaded area: [−277 ms, 442 ms]) between error and correct response waveforms.

**Figure 4 brainsci-13-00575-f004:**
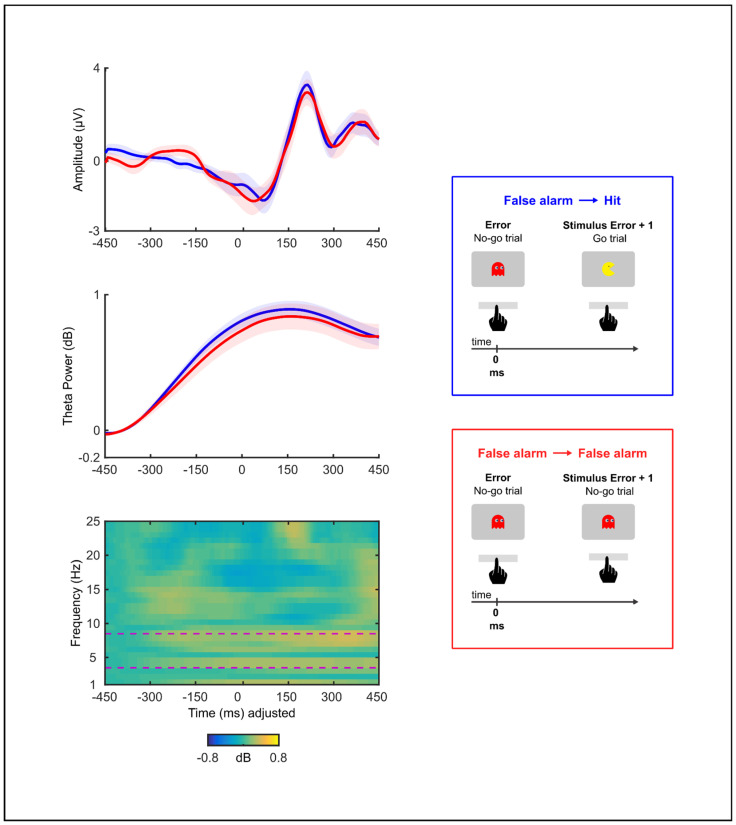
Grand averages of response-locked activity from single error trials and double error trials (F3/F4) (n = 45). The top graph shows time-domain amplitude. The middle graph shows temporal frequency powers in the theta band. The bottom graph shows temporal frequency powers after subtracting double error trials from single error trials in the frequency range from 1 to 25 Hz. Plots depict activity during the initial error. The button press occurred at 0 s. Permutation analysis revealed no significant differences (all *p* > 0.246) in amplitude [−50 ms, 150 ms] and theta power [−350 ms, 450 ms] of response-locked activity between double error and single error trials.

**Figure 5 brainsci-13-00575-f005:**
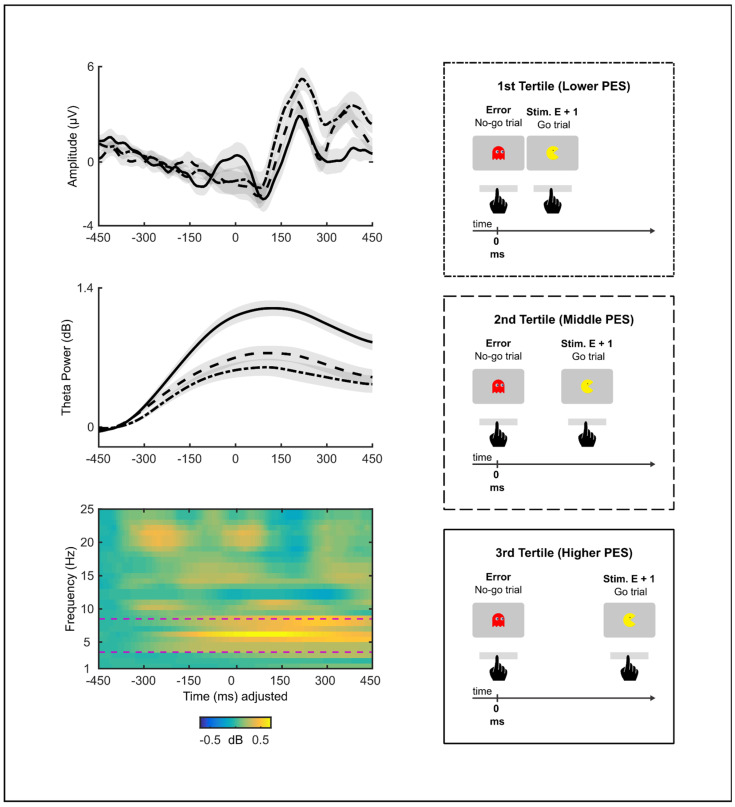
Grand averages of response-locked activity from error trials that preceded correct trials, which were divided into tertiles based on post-error slowing—PES (F3/F4) (n = 58). The top graph shows time-domain amplitude. The middle graph shows temporal frequency powers in the theta band. The bottom graph shows temporal frequency powers by subtracting the 1st tertile from the 3rd tertile in the frequency range from 4 to 25 Hz. The button press occurred at 0 s. There was a statistically significant difference between tertiles in the theta band [−350 ms, 450 ms] (*p* < 0.001), but not in the time-domain amplitude [−50 ms, 150 ms] (*p* = 0.772).

**Figure 6 brainsci-13-00575-f006:**
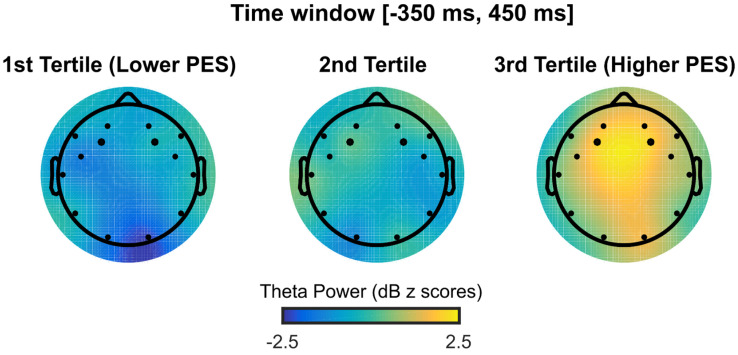
Topographic plots of theta power from error trials that preceded correct trials as a function of post-error slowing—PES tertiles throughout the respective analysis window. Black dots represent channel locations.

**Table 1 brainsci-13-00575-t001:** Demographic data of the sample.

Type of UBN Indicator	Proportion of the Sample
UBN ^1^	96.4%
UBNHEAD ^2^	9.6%
UBNEDU ^3^	2.7%
UBNHOUSE ^4^	26.2%
UBNWATER ^5^	0%
UBNBATHR ^6^	26.5%
UBNOVERC ^7^	14.3%
NEIGHBORHOOD ^8^	95.2%

^1^ Unsatisfied basic needs. ^2^ Homes with the head of family with incomplete high school and > 4 children in care. ^3^ Homes with school-age children (between 6 and 17 y of age) that were not attending school. ^4^ Inconvenient housing. ^5^ Homes with no access to drinking water. ^6^ Homes without bathroom, with shared bathroom, or with no discharge system. ^7^ Overcrowded homes. ^8^ Homes located in a vulnerable neighborhood.

**Table 2 brainsci-13-00575-t002:** Descriptive analysis for behavioral measures.

Measure	Median	Q25	Q75	n
Hits (%)	73.8	64.8	82.9	93
False Alarms (%)	36.1	26.1	50.9	93
Overall RT (ms)	561.1	518.8	607.4	93
Correct-trial RT (ms)	587.4	544.6	633.0	93
Error-trial RT (ms)	481.2	427.5	532.6	93
Post-correct slowing (ms)	10.3	−9.0	37.8	91
Post-error slowing (ms)	148.3	88.8	197.9	91
Post-correct accuracy (%)	83.5	76.1	88.8	91
Post-error accuracy (%)	83.6	76.2	89.6	91

## Data Availability

The datasets analyzed in the current study have been made available publicly at the GitHub repository Piettoetal_neural_processing_in_preschoolers and can be accessed at https://github.com/marcospietto/Piettoetal_error_processing_in_preschoolers, accessed on 26 March 2023.

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
