# Peer review of "EEG Dynamics of Error Processing and Associated Behavioral Adjustments in Preschool Children"

_brainsci, 2023, doi:10.3390/brainsci13040575_

Round 1
Reviewer 1 Report
The purpose of the present study was to investigate the neural activity associated with error processing (i.e., the ERN and frontal midline theta) and post-error behavioral adaptations, as well as to determine whether a relationship exists between them. Overall, the study provides an appropriate overview of the topic and cites some of the most relevant work in this field. The methods and analysis presented are compelling and well-presented, and the results are clearly displayed and well-integrated into the discussion.
However, there are a few comments that I think the authors should consider in the revision process.
Major comments
Although I appreciate the authors' efforts to move from an experimental setting to a more ecological context, it is not clear why they chose to collect data from Unsatisfied Basic Needs homes. The authors should clearly explain why they opted for this population and recognize that this population (i.e., children from low-socioeconomic status homes) may not be representative of the general preschool children population. This could limit the generalizability of the results.
The authors state that there is a lack of evidence regarding how neural responses to error predict subsequent behavioral adjustments during childhood. However, they also cite several studies investigating a possible relationship between behavioral adjustment and ERN and theta activity (e.g., Beatty et al., 2020; Cavanagh et al., 2015; Valadez et al., 2018). The authors should address this inconsistency (see for example paragraph at line 92-93 vs. 97-102).
Minor comments
Line 35. Please indicate a reference for this statement (“researchers generally use various tasks to study error processing (e.g., Flanker, Stroop, Simon, or Go/No-go tasks)”).
Line 139. Please explain the criteria used to exclude participants showing excessive artifacts in the EEG signal.
Line 143. Why did the authors administer the Sociodemographic Scale? If I understand correctly, it does not seem that the authors take the socioeconomic status of children into account in the analysis (and hypotheses in general).
Line 182. Based on what criteria can the task blocks range from 3 to 8?
Line 203. How many channels does the EEG system have?
Line 206. Please indicate the values of electrode impedances associated with yellow and green colors.
Lines 532-541. These aspects of the task should also be included in the Methods section.
Best.
Author Response
We thank the reviewer for the positive and constructive comments. Please, find the point-by-point response below.
Comment 1. Although I appreciate the authors' efforts to move from an experimental setting to a more ecological context, it is not clear why they chose to collect data from Unsatisfied Basic Needs homes. The authors should clearly explain why they opted for this population and recognize that this population (i.e., children from low-socioeconomic status homes) may not be representative of the general preschool children population. This could limit the generalizability of the results.
Response 1. Response 1. We appreciate the reviewer's comment. This study is one of the first to investigate the relation between the neural activity of error processing and subsequent compensatory behavior in children from UBN homes. Moreover, as these effects have not been explored in other child populations, expanding the sample to include different socioeconomic backgrounds is crucial since research has shown that the development of cognitive control could be affected by environmental factors [1–12]. Despite this, it is worth noting that there are no conclusive results concerning error monitoring [13–16]. Additionally, while certain studies have analyzed the correlation between the ERN and post-error behavior [17–20], none of them have utilized time-frequency measures, such as the frontal midline theta, to assess its association with PES and PEA. We have revised the introduction section providing additional information about the choice of a UBN sample (Page 3).
Comment 2. The authors state that there is a lack of evidence regarding how neural responses to error predict subsequent behavioral adjustments during childhood. However, they also cite several studies investigating a possible relationship between behavioral adjustment and ERN and theta activity (e.g., Beatty et al., 2020; Cavanagh et al., 2015; Valadez et al., 2018). The authors should address this inconsistency (see for example paragraph at line 92-93 vs. 97-102).
Response 2. We thank the Reviewer for this comment. The first paragraph (line: 92-93) only referred to studies conducted in children, while the second paragraph (line: 97-102) we reviewed studies in adults. For instance, the evidence regarding the relationship between behavioral adjustment and theta activity was solely obtained from adult studies. We have revised the second paragraph and added more specific information to clarify this (Page 3).
Comment 3. Line 35. Please indicate a reference for this statement (“researchers generally use various tasks to study error processing (e.g., Flanker, Stroop, Simon, or Go/No-go tasks)”).
Response 3. Thank you for pointing out this. We have included references to support the aforementioned statement in the current version of the manuscript (Page 2).
Comment 4. Line 139. Please explain the criteria used to exclude participants showing excessive artifacts in the EEG signal.
Response 4. We appreciate the Reviewer’s comment. The criterion to exclude participants with excessive artifacts was a low signal-to-noise ratio in the response-locked activity (without discriminating conditions). This evaluation was performed by comparing the variability between the baseline and the time interval of interest [-50, 350 ms].
Comment 5. Line 143. Why did the authors administer the Sociodemographic Scale? If I understand correctly, it does not seem that the authors take the socioeconomic status of children into account in the analysis (and hypotheses in general).
Response 5. The current work is part of a larger project in which the scale administration was fundamental to explore and confirm the presence of unsatisfied basic needs in our sample. In the current manuscript, we used the socioeconomic data from the scale only for descriptive purposes.
Comment 6. Line 182. Based on what criteria can the task blocks range from 3 to 8?
Response 6. Children could finish the task before completing all eight blocks if they were fatigued or chose not to proceed. Therefore, the number of blocks completed by the children varied between 3 and 8. We have provided this information in the current version of the manuscript (Page 5).
Comment 7. Line 203. How many channels does the EEG system have?
Response 7. The system has 14 saline-based electrodes that are positioned on the scalp according to the International 10-20 system. We have included this information in the current version of the manuscript (Page 6).
Comment 8. Line 206. Please indicate the values of electrode impedances associated with yellow and green colors.
Response 8. Emotiv does not provide information regarding the exact impedance level of its electrodes. Instead, the manufacturer provides a real-time contact quality measurement using a patented system and a color scale [emotiv contact quality]. Black indicates that the electrode is disconnected, Green indicates that the impedance is less than 5kΩ, yellow indicates an intermediate impedance value, and Red indicates that the electrode is unusable. They recommend maintaining the impedance level within the range indicated by the yellow to a green color scale, which is considered an appropriate level.
Comment 9. Lines 532-541. These aspects of the task should also be included in the Methods section.
Response 9. We briefly mention this inclusion criterion in the Methods section, and we expanded it to discuss its limitations in the Discussion further. In the current version of the manuscript, we have also mentioned these aspects in the Data analysis section (Page 7).
References
- St. John, A.M.; Finch, K.; Tarullo, A.R. Socioeconomic Status and Neural Processing of a Go/No-Go Task in Preschoolers: An Assessment of the P3b. Dev. Cogn. Neurosci. 2019, 38, 100677, doi:10.1016/j.dcn.2019.100677.
- Tomlinson, R.C.; Burt, S.A.; Waller, R.; Jonides, J.; Miller, A.L.; Gearhardt, A.N.; Peltier, S.J.; Klump, K.L.; Lumeng, J.C.; Hyde, L.W. Neighborhood Poverty Predicts Altered Neural and Behavioral Response Inhibition. NeuroImage 2020, 209, 116536, doi:10.1016/j.neuroimage.2020.116536.
- Rosen, M.L.; Sheridan, M.A.; Sambrook, K.A.; Meltzoff, A.N.; McLaughlin, K.A. Socioeconomic Disparities in Academic Achievement: A Multi-Modal Investigation of Neural Mechanisms in Children and Adolescents. NeuroImage 2018, 173, 298–310, doi:10.1016/j.neuroimage.2018.02.043.
- Sheridan, M.A.; Peverill, M.; Finn, A.S.; McLaughlin, K.A. Dimensions of Childhood Adversity Have Distinct Associations with Neural Systems Underlying Executive Functioning. Dev. Psychopathol. 2017, 29, 1777–1794, doi:10.1017/S0954579417001390.
- D’Angiulli, A.; Herdman, A.; Stapells, D.; Hertzman, C. Children’s Event-Related Potentials of Auditory Selective Attention Vary with Their Socioeconomic Status. Neuropsychology 2008, 22, 293–300, doi:10.1037/0894-4105.22.3.293.
- D’Angiulli, A.; Weinberg, J.; Oberlander, T.F.; Grunau, R.E.; Hertzman, C.; Maggi, S. Frontal EEG/ERP Correlates of Attentional Processes, Cortisol and Motivational States in Adolescents from Lower and Higher Socioeconomic Status. Front. Hum. Neurosci. 2012, 6, doi:10.3389/fnhum.2012.00306.
- Giuliano, R.J.; Karns, C.M.; Roos, L.E.; Bell, T.A.; Petersen, S.; Skowron, E.A.; Neville, H.J.; Pakulak, E. Effects of Early Adversity on Neural Mechanisms of Distractor Suppression Are Mediated by Sympathetic Nervous System Activity in Preschool-Aged Children. Dev. Psychol. 2018, 54, 1674–1686, doi:10.1037/dev0000499.
- Hampton Wray, A.; Stevens, C.; Pakulak, E.; Isbell, E.; Bell, T.; Neville, H. Development of Selective Attention in Preschool-Age Children from Lower Socioeconomic Status Backgrounds. Dev. Cogn. Neurosci. 2017, 26, 101–111, doi:10.1016/j.dcn.2017.06.006.
- Isbell, E.; Wray, A.H.; Neville, H.J. Individual Differences in Neural Mechanisms of Selective Auditory Attention in Preschoolers from Lower Socioeconomic Status Backgrounds: An Event-Related Potentials Study. Dev. Sci. 2016, 19, 865–880, doi:10.1111/desc.12334.
- Stevens, C.; Lauinger, B.; Neville, H. Differences in the Neural Mechanisms of Selective Attention in Children from Different Socioeconomic Backgrounds: An Event-Related Brain Potential Study. Dev. Sci. 2009, 12, 634–646, doi:10.1111/j.1467-7687.2009.00807.x.
- Kishiyama, M.M.; Boyce, W.T.; Jimenez, A.M.; Perry, L.M.; Knight, R.T. Socioeconomic Disparities Affect Prefrontal Function in Children. J. Cogn. Neurosci. 2009, 21, 1106–1115, doi:10.1162/jocn.2009.21101.
- Sullivan, E.F.; Xie, W.; Conte, S.; Richards, J.E.; Shama, T.; Haque, R.; Petri, W.A.; Nelson, C.A. Neural Correlates of Inhibitory Control and Associations with Cognitive Outcomes in Bangladeshi Children Exposed to Early Adversities. Dev. Sci. 2022, 25, doi:10.1111/desc.13245.
- Brooker, R.J. Maternal Behavior and Socioeconomic Status Predict Longitudinal Changes in Error-Related Negativity in Preschoolers. Child Dev. 2018, 89, 725–733, doi:10.1111/cdev.13066.
- Conejero, Á.; Guerra, S.; Abundis-Gutiérrez, A.; Rueda, M.R. Frontal Theta Activation Associated with Error Detection in Toddlers: Influence of Familial Socioeconomic Status. Dev. Sci. 2018, 21, e12494, doi:10.1111/desc.12494.
- Pietto, M.L.; Giovannetti, F.; Segretin, M.S.; Rueda, R.; Kamienkowski, J.E.; Lipina, S.J. Conflict-Related Brain Activity after Individualized Cognitive Training in Preschoolers from Poor Homes. J. Cogn. Enhanc. 2022, 6, 74–107, doi:10.1007/s41465-021-00223-6.
- Mistry-Patel, S.; Brooker, R.J. Associations between Error-Related Negativity and Childhood Anxiety Risk Differ Based on Socioeconomic Status. Dev. Psychol. 2022, doi:10.1037/dev0001461.
- Overbye, K.; Walhovd, K.B.; Paus, T.; Fjell, A.M.; Huster, R.J.; Tamnes, C.K. Error Processing in the Adolescent Brain: Age-Related Differences in Electrophysiology, Behavioral Adaptation, and Brain Morphology. Dev. Cogn. Neurosci. 2019, 38, 100665, doi:10.1016/j.dcn.2019.100665.
- Kang, K.; Alexander, N.; Wessel, J.R.; Wimberger, P.; Nitzsche, K.; Kirschbaum, C.; Li, S.-C. Neurocognitive Development of Novelty and Error Monitoring in Children and Adolescents. Sci. Rep. 2021, 11, 19844, doi:10.1038/s41598-021-99043-z.
- Wiersema, J.R.; van der Meere, J.J.; Roeyers, H. Developmental Changes in Error Monitoring: An Event-Related Potential Study. Neuropsychologia 2007, 45, 1649–1657, doi:10.1016/j.neuropsychologia.2007.01.004.
- Torpey, D.C.; Hajcak, G.; Kim, J.; Kujawa, A.; Klein, D.N. Electrocortical and Behavioral Measures of Response Monitoring in Young Children during a Go/No-Go Task. Dev. Psychobiol. 2012, 54, 139–150, doi:10.1002/dev.20590.

Reviewer 2 Report
The study examines behavioral adjustments and electrophysiological modulations associated with error processing in preschool children with low socio-economic status. To this aim, a Go/No-go task was used and EEG was recorded during task execution. The task was adapted to obtain a high number of errors. Data from 93 children were included in analyses, with a minimum of seven artifact-free error trials. Behavioral data confirmed the presence of post-error adjustments in responses (i.e., post-error slowing and higher post-error accuracy). Electrophysiological data showed larger frontal negativity (ERN) and higher frontal theta power after an error. The authors interpret data as reflecting adaptive mechanisms.
The attempt of the authors to investigate this important control function in early childhood is noteworthy. The study hypotheses have been founded on previous literature and clearly described. The results confirm and extend previous findings. However, some major points should be addressed, as detailed below.
Overall, it is unclear the reasons why children of low SES should differ in error monitoring and behavioral adjustments from the rest of the children. The authors should better describe the rationale for it.
If the aim of the study was to test whether and how living in “Unsatisfied Basic Needs” homes affects error processing and the following adjustments, a group of children not living in UBN homes should have been tested as well. If poor living contexts affect error processing and following behavioral adjustments I would expect an association, likely linear, between the number of UBN indicators. Could the authors test it?
Participants
Please indicate how many children were included in the final analyses.
I wonder how the participants’ selection has been conducted. How were developmental disorders assessed? Were intellectual disabilities assessed as well? What do anthropometric standards include?
In the literature, a minimum of seven trials has been set for estimating error-related potentials. However, this threshold has been set on adults (e.g., Olvet & Hajcak, 2009) and we know that signal-to-noise ratio in children is much higher. The authors should take into account this limitation. Moreover, the range of the number of trials included in the analyses, together with the median (line 140), show be declared. On average, how many hits and false alarms were included in the analysis? Was the same for behavioral and EEG?
Task
In the task, the authors included four types of no-go stimuli and only one type of go stimuli (line 176). Please, explain the reason of this choice. Was the no-go stimulus the same within one block?
No-go trials could be successive? From Figure 4 it appears that a no-go trial could follow another no-go trial. Please specify.
Does the stimulus disappear upon button press? This is an important point given that the EEG evoked by the button press is examined. Moreover, there is an incongruency between the timing information provided in the text and the information provided in Figure 1. Did the stimulus last 400 or 800 ms?
Behavioral analyses
I found misleading lines 193-197. Please clarify and rephrase. Do you mean that RTs after correct no-go trials were contrasted to 20 randomly selected RTs after correct hits? What does it mean that for accuracy you selected “post-trials” associated with hits and false alarms and computed the percentage of responses associated with hits? Was the computation as follows: Number of hits after correct go trials + Number of hits after false alarms trials/number of hits? Why Post-correct accuracy was considered?
I wonder whether the authors included responses faster than 80-100 ms, which are usually considered anticipated responses and excluded from the analysis.
EEG
How large was the threshold for impedance during EEG recording? Where were placed electrodes, physical reference and ground electrodes especially? I am not sure that the average reference is an optimal choice when the number of electrodes is low.
It is unclear which was the baseline interval for ERSP analysis, [-450 ms - 50 ms] (line 231) or [-450 ms - 350 ms] (line 241)?
Non-parametric permutation t-tests were performed along [-50 ms - 150 ms] and [-350 ms - 450 ms] time windows, for ERP and ERSP, respectively. Weren’t these intervals included as the baseline or as the interval of interest in the ERSP analysis (lines 216 and 241)? Why the [-350 ms - 450 ms] time window was chosen for the ERPS analysis? One hundred milliseconds for analyzing theta power perhaps is not optimal given that one complete cycle of theta rhythm takes 250 ms.
Overall, paragraphs on planned analyses are hard to follow. I would suggest arranging EEG analysis according to the hypotheses.
What does it mean “multiple slow-down levels”? (line 281)
I cannot understand the aim of the “Response-Stimulus Interval Analysis”. What interval has been exactly examined here? Which is the “next task-relevant stimulus”? Please rephrase and better explain the rationale of this point.
F3 and F4 are not midline electrode positions and source analysis cannot be performed, therefore, it is incorrect to name “frontal midline theta” the electrophysiological index targeted in analyses (e.g., line 404)
Results
The grey shadows in Figure 3 are misleading. It should be acknowledged time intervals chosen a priori and that significant in the analysis.
Discussion
I cannot follow the reasoning in lines 492-494.
The authors should acknowledge the limitation of using the Emotiv system and very few electrodes. This montage cannot properly detect electrophysiological activity all over the scalp.
I wonder whether the association of PES and PEA reflects the speed-accuracy trade-off only, which is likely high in preschool children.
The link to the GitHub repository does not work
When ethical approval was obtained?
Minor
I am not an English native speaker, but I think that English needs to be improved.
Line 35: Insert at least the first name of the author in citation n. 3.
Please, remove the acronym PEA or replace “Post-error accuracy” with the acronym from line 195.
Spell out RSI line 286.
Table 1. Please explain what “dependents” refer to.
Please replace “error commission” with “error of commission”.
Author Response
We thank the reviewer for his thoughtful comments. Despite the brief period of time allowed for the revision, we could answer all the suggestions, including revising the writing by a native professional English editor. Please find the point-by-point responses below.
Comment 1. Overall, it is unclear the reasons why children of low SES should differ in error monitoring and behavioral adjustments from the rest of the children. The authors should better describe the rationale for it.
Response 1. We appreciate the reviewer's suggestion. Although we did not specifically assess differences between low and high SES in this study, other studies showed differences in tasks with demands of cognitive control [1–12]. In particular, there were four studies exploring error monitoring processes. They showed that the neural dynamics of error processing might be reduced in amplitude and power in toddlers from low SES contexts compared to their peers from high SES [13]. Also, the neural activity associated with errors in preschool children living in lower SES could have different developmental trajectories [14] and associations with self-regulatory behaviors [15] compared with their counterparts living in higher SES contexts. Furthermore, the neural activity related to error processing of preschool children from low SES homes may be modified in a developmental trajectory following a cognitive intervention [16]. We have revised the introduction section providing additional information about the rationale of studying children from low-SES backgrounds and the supporting evidence related to error monitoring (Page 3).
Comment 2. If the aim of the study was to test whether and how living in “Unsatisfied Basic Needs” homes affects error processing and the following adjustments, a group of children not living in UBN homes should have been tested as well. If poor living contexts affect error processing and following behavioral adjustments I would expect an association, likely linear, between the number of UBN indicators. Could the authors test it?
Response 2. We agreed with the reviewer. Nevertheless, the study aimed not to test whether and how living in “Unsatisfied Basic Needs” homes affects error processing and the following behavioral adjustments, but a necessary first step in that direction. The present study is part of a larger project to evaluate self-regulatory processes in children living in UBN homes. As part of that project, the current study investigated how error processing occurs in children living in UBN homes. Specifically, we sought to examine the neural and behavioral aspects of error processing and the interplay between them to characterize error monitoring in this population. The analysis suggested by the reviewer is very interesting. However, we cannot conduct this analysis as our sample exhibits limited socioeconomic variability and sparse distribution across various UBN indicators. Already planned follow-up studies will go in that direction.
Comment 3. Please indicate how many children were included in the final analyses.
Response 3. We appreciate the reviewer's suggestion. The sample size varied across the different analyses, ranging from 58 to 93 subjects. The reason for the sample variation in the analyses was that the cutoff of 7 trials was not met. In the current manuscript version, we have included details about the sample size for each analysis in both the results section and the figure captions (Pages 10-14).
Comment 4. I wonder how the participants’ selection has been conducted. How were developmental disorders assessed? Were intellectual disabilities assessed as well? What do anthropometric standards include?
Response 4. The anthropometric measurements included head size, weight, and height measurements. While we did not conduct direct assessments for developmental disorders, we obtained information from school authorities, teachers, and parents regarding any previous occurrences of developmental disorders in the children's medical history. Also, when we requested information about the developmental disorders history, we investigated the presence of intellectual disabilities in the children.
All authorized children participate in the activities. Although, in the pre-analytical phase, children with developmental disorders reported by school authorities, teachers, and parents were excluded from all the analyses (N = 3).
Comment 5. (a) In the literature, a minimum of seven trials has been set for estimating error-related potentials. However, this threshold has been set on adults (e.g., Olvet & Hajcak, 2009) and we know that signal-to-noise ratio in children is much higher. The authors should take into account this limitation.
Response 5. (a) We agree with the reviewer's comment and acknowledge that the implemented threshold is limited because it has been tested only on adults. Accordingly, we have mentioned this limitation in the discussion section (Page 18).
(b) Moreover, the range of the number of trials included in the analyses, together with the median (line 140), show be declared.
Response 5. (b) We have included the following table containing information on the number of trials conducted for each analysis as supplemental material 3.
Table S3 1. Number of trials for each variable used in the analysis |
|||
Behavioral measures |
Median |
Min |
Max |
Correct-trial RT (ms) |
326 |
46 |
503 |
Error-trial RT (ms) |
67 |
8 |
177 |
Post-correct slowing (ms) |
39 |
7 |
129 |
Post-error slowing (ms) |
39 |
7 |
129 |
Post-correct accuracy (%) |
39 |
7 |
129 |
Post-error accuracy (%) |
39 |
7 |
129 |
Neural measures |
Median |
Min |
Max |
Correct-related activity |
311 |
45 |
497 |
Error-related activity |
60 |
8 |
169 |
Error-related activity before a correct response |
45 |
11 |
84 |
Error-related activity before an error response |
12 |
7 |
39 |
Error-related activity before a correct response 1st tertile |
15 |
7 |
29 |
Error-related activity before a correct response 2nd tertile |
14 |
7 |
27 |
Error-related activity before a correct response 3rd tertile |
14 |
8 |
34 |
(c) On average, how many hits and false alarms were included in the analysis? Was the same for behavioral and EEG?
Response 5. (c) The mean number of hits and false alarms were 308 and 66, respectively. However, there was a difference in the number of trials conducted for behavioral and EEG measures. Specifically, the number of hits was 314 for behavior and 302 for EEG, while false alarms were 68 for behavior and 64 for EEG.
Comment 6. In the task, the authors included four types of no-go stimuli and only one type of go stimuli (line 176). Please, explain the reason of this choice. Was the no-go stimulus the same within one block?
Response 6. We appreciate the reviewer's comments. To ensure consistency within each block, we employed the same type of No-go stimulus (either Ghost or Pig). However, the same No-go stimulus could appear in four different colors within a block. We adjusted the experiment parameters to prevent fatigue and demotivation among the children participating in the study, thereby increasing the probability of completing the task.
Comment 7. No-go trials could be successive? From Figure 4 it appears that a no-go trial could follow another no-go trial. Please specify.
Response 7: Yes, the No-go trials could be successive because the distribution of Go and No-go trials was randomized across all task blocks.
Comment 8. Does the stimulus disappear upon button press? This is an important point given that the EEG evoked by the button press is examined.
Response 8. No, the stimulus did not disappear following the button press. We included this information in the task section of the current version of the manuscript (Page 5).
Comment 9: Moreover, there is an incongruency between the timing information provided in the text and the information provided in Figure 1. Did the stimulus last 400 or 800 ms?
Response 9: We thank the reviewer for their careful reading. We have now rectified an error in Figure 1, where the stimulus time was incorrectly depicted as 800 ms. The stimulus time was, in fact, 400 ms, while the fixation time was 800 ms (Page 6).
Comment 10. I found misleading lines 193-197. Please clarify and rephrase. Do you mean that RTs after correct no-go trials were contrasted to 20 randomly selected RTs after correct hits? What does it mean that for accuracy you selected “post-trials” associated with hits and false alarms and computed the percentage of responses associated with hits? Was the computation as follows: Number of hits after correct go trials + Number of hits after false alarms trials/number of hits? Why Post-correct accuracy was considered?
Response 10. Post-correct measures (i.e., Post-correct RT, Post-correct slowing, and Post-correct Accuracy) were matched to Post-error measures (i.e., Post-error RT, Post-error slowing, and Post-error accuracy). We selected the same number of trials for error and correct responses to control for differences in RT due to regression toward the mean. Post-correct measures refer to responses that occur after a hit, i.e., following the presentation of a Go stimulus with the subsequent button press. In contrast, Post-error measures refer to responses that occur after a false alarm, i.e., following the presentation of a No-go stimulus with an associated response. Based on this, Post-correct accuracy and Post-error accuracy were calculated as the percentage of correct responses that occurred after a hit and false alarm, respectively. We considered both RTs associated with hits and false alarms when calculating the percentage. We used the Post-correct accuracy as a reference point to assess whether error processing significantly improves performance compared to processing a correct response [17]. By comparing Post-correct accuracy with Post-error accuracy, we could evaluate the impact of error processing on subsequent performance. We have clarified this point in the task section (Page 5).
Comment 11. I wonder whether the authors included responses faster than 80-100 ms, which are usually considered anticipated responses and excluded from the analysis.
Response 11. We appreciate the reviewer’s comment. Correct responses (i.e., hits) were only considered valid if they were made after 200 ms, while error responses (i.e., false alarms) were not subject to such restrictions. Thus these could involve responses faster than 200 ms. In this sense, we classified impulsive responses as errors and subsequently analyzed the neural activity and behavioral adjustments following them. Our findings revealed that false alarms with faster responses were linked to higher theta activity and greater post-error slowing (PES) before a correct response, suggesting that premature responses (including impulsive ones) facilitated an increment in subsequent error monitoring and adjustment processes. This statement is based on the results reported in the RSI (Response-stimulus interval) section and Figures 2 and 5. The PES tertiles that were calculated to analyze both behavior and neural activity after an error showed differences in the RSI. Specifically, we found that the larger PES tertiles (i.e., those associated with greater accuracy in behavior and higher theta in neural activity) displayed a longer interval between the response made to the No-go stimulus and the subsequent appearance of the next stimulus in the task. Given that the time between the presentation of two stimuli in the task was fixed, a longer RSI interval indicated that the response to the first stimulus (i.e., No-go) was made earlier or more prematurely, resulting in a longer interval before the next stimulus appeared.
Comment 12. (a) How large was the threshold for impedance during EEG recording? (b)Where were placed electrodes, physical reference and ground electrodes especially? (c)I am not sure that the average reference is an optimal choice when the number of electrodes is low.
Response 12: We thank the reviewer’s questions.
(a) Emotiv does not provide information regarding the exact impedance level of its electrodes. Instead, the manufacturer provides a real-time contact quality measurement using a patented system and a color scale [emotiv contact quality]. Black indicates that the electrode is disconnected, Green indicates that the impedance is less than 5kΩ, Yellow indicates an intermediate impedance value, and Red indicates that the electrode is unusable. They recommend maintaining the impedance level within the range indicated by the Yellow to a Green color scale, which is considered an appropriate level.
(b) CMS and DRL references were located at P3 and P4 (not available for the analysis), respectively.
(c) The Emotiv device has the limitation that the electrodes are fixed in place and not uniformly distributed. This means that it is not possible to use mastoids or an average reference. Given that our study only intended to focus on analyzing the activation patterns specifically on the F3/F4 electrodes, we created a pseudo-average reference across symmetrically distributed electrodes around the scalp. We have included this information in the Event-related Potentials (ERPs) section (Page 6).
Comment 13. It is unclear which was the baseline interval for ERSP analysis, [-450 ms - 50 ms] (line 231) or [-450 ms - 350 ms] (line 241)?
Response 13. We thank the reviewer for pointing this out. The baseline interval used for ERSP analysis was [-450 ms, -350 ms], while the interval [-450 ms, -50 ms] was utilized for ERP analysis. We apologize for repeating this information in the "Event-related Spectral Perturbations (ERSPs)" section and have now removed the duplicate information (Pages 6-7).
Comment 14. Non-parametric permutation t-tests were performed along [-50 ms - 150 ms] and [-350 ms - 450 ms] time windows, for ERP and ERSP, respectively. Weren’t these intervals included as the baseline or as the interval of interest in the ERSP analysis (lines 216 and 241)? Why the [-350 ms - 450 ms] time window was chosen for the ERPS analysis? One hundred milliseconds for analyzing theta power perhaps is not optimal given that one complete cycle of theta rhythm takes 250 ms.
Response 14. Non-parametric permutation t-tests were conducted from -50 ms to +150 ms for ERPs and from -350 ms to +450 ms for ERSPs. Therefore, the ERP analyses consisted of a 200 ms time window, whereas the ERSP analyses utilized an 800 ms time window. To clarify this point, we have revised the writing of the time intervals by adding a comma, such that they are now presented as [-50, 150] and [-350, 450] (Page 8).
Comment 15: Overall, paragraphs on planned analyses are hard to follow. I would suggest arranging EEG analysis according to the hypotheses.
Response 15. We thank the reviewer's suggestion. We have reformulated the data analysis section according to the hypotheses (Pages 7-8).
Comment 16: What does it mean “multiple slow-down levels”? (line 281)
Response 16. The sentence refers to groups based on the post-error slowing (PES) created through the computation of tertiles. We have replaced this sentence with "multiple PES levels" (Page 8).
Comment 17. I cannot understand the aim of the “Response-Stimulus Interval Analysis”. What interval has been exactly examined here? Which is the “next task-relevant stimulus”? Please rephrase and better explain the rationale of this point.
Response 17. Response-Stimulus Interval (RSI) refers to the amount of time between the participant’s response to a stimulus and the presentation of the next task stimulus. Research findings have indicated that the length of the RSI following an error response may influence PES and PEA in both adults [18,19] and children [20]. We computed it to examine whether varying PES levels corresponded to different RSI intervals. We computed it to examine whether varying PES levels corresponded to different RSI intervals. Notably, higher PES values were linked to enhanced accuracy after an error and higher power of theta oscillations (prior to a correct response). We have revised this section and included additional details to clarify the reasoning behind this analysis (Page 8).
Comment 18. F3 and F4 are not midline electrode positions and source analysis cannot be performed, therefore, it is incorrect to name “frontal midline theta” the electrophysiological index targeted in analyses (e.g., line 404)
We acknowledge the reviewer's comment and agree that the term could be misleading. We introduced it because we analyzed the average between F3/F4, which is the best we have to link with the literature. But, we have revised the manuscript accordingly and removed the term 'midline' from the expression referring to theta.
Comment 19. The grey shadows in Figure 3 are misleading. It should be acknowledged time intervals chosen a priori and that significant in the analysis.
Response 19. In the figure caption, we have now provided the time interval selected a priori and the significant time interval.
Comment 20. I cannot follow the reasoning in lines 492-494.
Response 20. We have rephrased the entire paragraph in page 17 as follows:
“One possible explanation for this pattern of results is that the highest PES tertile was more likely to reflect an adaptive or genuine recovery from an error compared with the lowest and middle PES tertiles. Contrary to the lowest and middle PES tertiles, the highest PES also involved longer RSI [21], that is, an additional time to process errors correctly before subsequent presentation of trial stimuli. It has been hypothesized that controlled processes are implemented chronologically later than automatic post-error processes when there is a long time between the error response and the subsequent presentation of trial stimuli [19]. Therefore, recovery from error at greater PES and longer RSI could have captured deliberated and controlled processes that enabled a strategic compensation of behavior. Conversely, a correct response that followed an error at smaller PES and shorter RSI could have indicated no real behavior correction, but rather a perseverative automatic motor response from the previous error trial.”
Comment 21. The authors should acknowledge the limitation of using the Emotiv system and very few electrodes. This montage cannot properly detect electrophysiological activity all over the scalp.
Response 21. We have included the following sentence in the discussion section (Page 19):
“Nevertheless, we recognize a constraint in using mobile EEG technology with a limited number of electrodes because this montage may not capture electrophysiological signals accurately across the entire scalp.”
Comment 22. I wonder whether the association of PES and PEA reflects the speed-accuracy trade-off only, which is likely high in preschool children.
Response 22. We think the speed-accuracy trade-off was most involved in the 3rd tertile compared with the 2nd and 1st tertiles. As the children were given more time to process errors, indicated by a larger RSI, we observed a corresponding increase in PEA. This leads us to believe that in the 3rd tertile, correct responses following errors were more likely due to adaptive processes [21]. However, in the 1st tertile of PES, where children had less time to process errors, correct responses following errors were more likely to indicate a lack of genuine error recovery (i.e., a perseverative automatic motor response from the previous error trial).
Comment 23. The link to the GitHub repository does not work.
Response 23. We apologize for not specifying that the link will work upon acceptance of the manuscript for publication. It is now publicly available.
Comment 24. When ethical approval was obtained?
Response 24. Ethical approvals for protocol nos. 682 and 961 were obtained in 2011 and 2015, respectively.
Comment 25. I am not an English native speaker, but I think that English needs to be improved.
Response 25. The new version of the manuscript has been reviewed by a native English professional editor. We hope that now it is clearer.
Comment 26. Line 35: Insert at least the first name of the author in citation n. 3.
Response 26. We have added the name for that citation.
Comment 27. Please, remove the acronym PEA or replace “Post-error accuracy” with the acronym from line 195.
Response 27. We have replaced “Post-error accuracy” with the acronym PEA.
Comment 28. Spell out RSI line 286.
Response 28. Thank you for this remark.
Comment 29. Table 1. Please explain what “dependents” refer to.
Response 29. We have modified the sentence and reformulated it as “Homes with the head of the family with incomplete high school and > 4 children in care.”
Comment 30. Please replace “error commission” with “error of commission”.
Response 30. We have replaced “error commission” with “an error”.
References
- St. John, A.M.; Finch, K.; Tarullo, A.R. Socioeconomic Status and Neural Processing of a Go/No-Go Task in Preschoolers: An Assessment of the P3b. Dev. Cogn. Neurosci. 2019, 38, 100677, doi:10.1016/j.dcn.2019.100677.
- Sullivan, E.F.; Xie, W.; Conte, S.; Richards, J.E.; Shama, T.; Haque, R.; Petri, W.A.; Nelson, C.A. Neural Correlates of Inhibitory Control and Associations with Cognitive Outcomes in Bangladeshi Children Exposed to Early Adversities. Dev. Sci. 2022, 25, doi:10.1111/desc.13245.
- Tomlinson, R.C.; Burt, S.A.; Waller, R.; Jonides, J.; Miller, A.L.; Gearhardt, A.N.; Peltier, S.J.; Klump, K.L.; Lumeng, J.C.; Hyde, L.W. Neighborhood Poverty Predicts Altered Neural and Behavioral Response Inhibition. NeuroImage 2020, 209, 116536, doi:10.1016/j.neuroimage.2020.116536.
- Rosen, M.L.; Sheridan, M.A.; Sambrook, K.A.; Meltzoff, A.N.; McLaughlin, K.A. Socioeconomic Disparities in Academic Achievement: A Multi-Modal Investigation of Neural Mechanisms in Children and Adolescents. NeuroImage 2018, 173, 298–310, doi:10.1016/j.neuroimage.2018.02.043.
- Sheridan, M.A.; Peverill, M.; Finn, A.S.; McLaughlin, K.A. Dimensions of Childhood Adversity Have Distinct Associations with Neural Systems Underlying Executive Functioning. Dev. Psychopathol. 2017, 29, 1777–1794, doi:10.1017/S0954579417001390.
- D’Angiulli, A.; Herdman, A.; Stapells, D.; Hertzman, C. Children’s Event-Related Potentials of Auditory Selective Attention Vary with Their Socioeconomic Status. Neuropsychology 2008, 22, 293–300, doi:10.1037/0894-4105.22.3.293.
- D’Angiulli, A.; Weinberg, J.; Oberlander, T.F.; Grunau, R.E.; Hertzman, C.; Maggi, S. Frontal EEG/ERP Correlates of Attentional Processes, Cortisol and Motivational States in Adolescents from Lower and Higher Socioeconomic Status. Front. Hum. Neurosci. 2012, 6, doi:10.3389/fnhum.2012.00306.
- Giuliano, R.J.; Karns, C.M.; Roos, L.E.; Bell, T.A.; Petersen, S.; Skowron, E.A.; Neville, H.J.; Pakulak, E. Effects of Early Adversity on Neural Mechanisms of Distractor Suppression Are Mediated by Sympathetic Nervous System Activity in Preschool-Aged Children. Dev. Psychol. 2018, 54, 1674–1686, doi:10.1037/dev0000499.
- Hampton Wray, A.; Stevens, C.; Pakulak, E.; Isbell, E.; Bell, T.; Neville, H. Development of Selective Attention in Preschool-Age Children from Lower Socioeconomic Status Backgrounds. Dev. Cogn. Neurosci. 2017, 26, 101–111, doi:10.1016/j.dcn.2017.06.006.
- Isbell, E.; Wray, A.H.; Neville, H.J. Individual Differences in Neural Mechanisms of Selective Auditory Attention in Preschoolers from Lower Socioeconomic Status Backgrounds: An Event-Related Potentials Study. Dev. Sci. 2016, 19, 865–880, doi:10.1111/desc.12334.
- Stevens, C.; Lauinger, B.; Neville, H. Differences in the Neural Mechanisms of Selective Attention in Children from Different Socioeconomic Backgrounds: An Event-Related Brain Potential Study. Dev. Sci. 2009, 12, 634–646, doi:10.1111/j.1467-7687.2009.00807.x.
- Kishiyama, M.M.; Boyce, W.T.; Jimenez, A.M.; Perry, L.M.; Knight, R.T. Socioeconomic Disparities Affect Prefrontal Function in Children. J. Cogn. Neurosci. 2009, 21, 1106–1115, doi:10.1162/jocn.2009.21101.
- Conejero, Á.; Guerra, S.; Abundis-Gutiérrez, A.; Rueda, M.R. Frontal Theta Activation Associated with Error Detection in Toddlers: Influence of Familial Socioeconomic Status. Dev. Sci. 2018, 21, e12494, doi:10.1111/desc.12494.
- Brooker, R.J. Maternal Behavior and Socioeconomic Status Predict Longitudinal Changes in Error-Related Negativity in Preschoolers. Child Dev. 2018, 89, 725–733, doi:10.1111/cdev.13066.
- Mistry-Patel, S.; Brooker, R.J. Associations between Error-Related Negativity and Childhood Anxiety Risk Differ Based on Socioeconomic Status. Dev. Psychol. 2022, doi:10.1037/dev0001461.
- Pietto, M.L.; Giovannetti, F.; Segretin, M.S.; Rueda, R.; Kamienkowski, J.E.; Lipina, S.J. Conflict-Related Brain Activity after Individualized Cognitive Training in Preschoolers from Poor Homes. J. Cogn. Enhanc. 2022, 6, 74–107, doi:10.1007/s41465-021-00223-6.
- Valadez, E.A.; Simons, R.F. The Power of Frontal Midline Theta and Post-Error Slowing to Predict Performance Recovery: Evidence for Compensatory Mechanisms. Psychophysiology 2018, 55, e13010, doi:10.1111/psyp.13010.
- Beatty, P.J.; Buzzell, G.A.; Roberts, D.M.; McDonald, C.G. Contrasting Time and Frequency Domains: ERN and Induced Theta Oscillations Differentially Predict Post-Error Behavior. Cogn. Affect. Behav. Neurosci. 2020, 20, 636–647, doi:10.3758/s13415-020-00792-7.
- Jentzsch, I.; Dudschig, C. Short Article: Why Do We Slow down after an Error? Mechanisms Underlying the Effects of Posterror Slowing. Q. J. Exp. Psychol. 2009, 62, 209–218, doi:10.1080/17470210802240655.
- Smulders, S.F.A.; Soetens, E.; van der Molen, M.W. What Happens When Children Encounter an Error? Brain Cogn. 2016, 104, 34–47, doi:10.1016/j.bandc.2016.02.004.
- Wessel, J.R. An Adaptive Orienting Theory of Error Processing. Psychophysiology 2018, 55, e13041, doi:10.1111/psyp.13041.
